# Is normalization indispensable
# for training deep neural networks?

**Jie Shao**[1*], **Kai Hu**[2*], **Changhu Wang**[3], **Xiangyang Xue**[1], **Bhiksha Raj**[2]
[1]Fudan University, Shanghai, China
[2]Carnegie Mellon University, Pittsburgh, PA
[3]Byte Dance AI Lab, Shanghai, China
{kaihu, bhiksha}@cs.cmu.edu, {shaojie, xyxue}@fudan.edu.cn,
wangchanghu@bytedance.com

## Abstract

Normalization operations are widely used to train deep neural networks, and they can improve both convergence and generalization in most tasks. The theories for normalization's effectiveness and new forms of normalization have always been hot topics in research. To better understand normalization, one question can be whether normalization is indispensable for training deep neural networks? In this paper, we analyze what would happen when normalization layers are removed from the networks, and show how to train deep neural networks without normalization layers and **without performance degradation**. Our proposed method can achieve the same or even slightly better performance in a variety of tasks: image classification in ImageNet, object detection and segmentation in MS-COCO, video classification in Kinetics, and machine translation in WMT English-German, etc. Our study may help better understand the role of normalization layers and can be a competitive alternative to normalization layers. Codes are available at `https://github.com/hukkai/rescaling`.

## 1   Introduction

Deep neural networks have greatly advanced the benchmarks in many artificial intelligence applications, such as image recognition [19], speech recognition [1], and natural language processing [32], etc. However, training effective deep neural networks is often non-trivial, and beset by many problems, one of most annoying of which might be that of vanishing or exploding gradients, which directly relates to the problem of vanishing or exploding *variance* of a signal (*i.e.* an input) as it passes through the network [13]. Batch Normalization (BN) [17] greatly mitigates this problem. Since the introduction of BN, several variants have been proposed that apply the underlying principle to a wider range of tasks: Layer Normalization for recurrent neural networks [2], Instance Normalization (IN) [33] for stylization, Group Normalization (GN) [36] for small-batch training, etc. Normalization operations are, by now, default components of the state of the art in many tasks.

Despite the popularity and success of normalization, the theory behind the effectiveness of such operations in neural networks is not yet fully understood. The original motivation provided in [17] is that BN can reduce internal covariate shift. Other studies [29, 22, 4, 38] also show several other advantages provided by normalization in terms of loss landscape, regularization, etc. However, most theoretical analyses require assumptions about the data/feature distribution, and the developed theories are only evaluated on small datasets. Analyzing the optimization and generalization of normalization in deep networks is still an open problem.

---

[*]Equal contribution.

Since the effectiveness of normalization is still almost a black box, a natural question arises: is it indispensable for training deep neural networks? There are, actually, two parts to this question: 1) Can training of non-normalized models be stable? 2) Can we train non-normalized models and achieve the same performance as the corresponding normalized models? The answer to the first question is clear. The simplest solution is to train the network with very a small learning rate (although this may lead to bad local minimum). A better solution is Fix-up initialization [40]. With this careful initialization, residual networks can be trained with large learning rate and limited performance degradation.

This paper focuses on the second question. By analyzing what would happen when normalization layers are removed from the networks, we show how to train deep neural networks without normalization layers and **without performance degradation**.

We focus on the residual network (ResNet) [14], since residual learning tends to have better performance when the network is very deep. Our method starts with solving the vanishing/exploding variance problem (which is the main problem solved by normalization). Note that good initialization [13] can mitigate this problem in a conventional non-residual network. However, the problem of explosion returns with residual connections of the form $\boldsymbol{y}_i = \boldsymbol{x}_i + \mathcal{F}(\boldsymbol{x}_i)$. We propose RescaleNet to handle this problem, with the new formulation $\boldsymbol{y}_i = \alpha_i \boldsymbol{x}_i + \beta_i \mathcal{F}(\boldsymbol{x}_i)$ where $\alpha_i$ and $\beta_i$ are carefully selected constants. Concomitantly, we propose a novel network bias setting to compensate for the common problem of "dead" neurons that arise in un-normalized networks.

We validate our method on a wide range of tasks. On ImageNet, our un-normalized RescaleNet models can achieve the same or slightly better performance than the corresponding normalized models (ResNet, ResNext) with the same training settings. Our un-normalized RescaleNet variant on ResNet50 has 0.3% lower error than its BN/GN ResNet50 counterpart. Our method can also apply to conventional non-residual networks. Our 19 layer VGG [30] model without normalization achieves a top-1 validation error rate of 25.0%, which is 2.6% lower than PyTorch's pre-trained model [26]. Our method also shows consistent improvement on Mask R-CNN for COCO object detection and segmentation [20], 3D convolutional networks for Kinetics video classification [18], and deep transformers for WMT English-German machine translation [34]. In cases where normalization operations may cause problems, our method can be a competitive alternative.

## 2 Related Work

**Normalization** is believed to be essential for training deep neural networks. Batch Normalization (BN) [17] enables training with a large learning rate and largely solves the gradient explosion/vanishing problem. Layer Normalization (LN) [2] computes normalization statistics from all summed inputs to the neurons in a layer, and can stabilize the hidden state dynamics in a recurrent network. Instance Normalization (IN) [33] computes the statistics for each channel in each datum independently. Group Normalization [36] computes normalization statistics over groups of channels, making it effective for small batch settings. Switchable Normalization proposes a learning-to-normalize framework that switches between BN, LN, and IN. Weight normalization [28] is a reparameterization of the weights to separate the direction and length of weights.

**Good Initialization** is essential for non-normalized networks. Xavier Initialization [11] estimates the standard deviation of initial parameter values on the basis of the number of input and output channels in a layer. He initialization [13] extends the formula to the ReLU activation, making it possible to train deeper networks. However, these methods do not work for residual networks [15]. Goyal *et al.* [12] finds initializing the residual branches by zero can ease optimization. Two recent studies, Fixup Init[40] and SkipInit [7] are based on this observation. Our method is different from these studies: it does not benefit from zero initialization. Besides initialization, scaling the hidden layers is also commonly used to control the activation magnitude [9, 3].

## 3 Preliminaries

We begin by briefly discussing two key challenges in training ResNets, to motivate our work. At its core, a ResNet computes its output through incremental additive connections, obtained from a sequence of *residual* blocks. Let $\boldsymbol{x}_0$ be the input to the net and $\boldsymbol{x}_l$ be the output after the $l^{\text{th}}$ residual connection. Let $\mathcal{F}_l$ be the corresponding residual block. The $l^{\text{th}}$ residual connection is formulated as $\boldsymbol{x}_l = \boldsymbol{x}_{l-1} + \mathcal{F}_l(\boldsymbol{x}_{l-1})$. Each residual block $\mathcal{F}_l$ in turn is a multilayer network, typically a

convolutional network or even an MLP, with weights (filters) $\boldsymbol{W}_{l,k}$ and biases $\boldsymbol{b}_{l,k}$, $k = 1 \cdots K_l$ (where $K_l$ is the depth of $\mathcal{F}_l$). Activations are generally ReLUs, as will be assumed in this paper.

ResNets typically also include normalization layers within and between residual blocks, however since our objective is to develop a framework that does not require them, we will assume they are not present. We will refer to ResNets that do not include normalization layers as "plain" ResNets.

## 3.1 The problem of exploding variance

As previously reported in several papers, random initialization of weights within a multi-layer network can result in unstable (exploding or vanishing) variances of the activations with increasing depth, and a variety of initialization strategies have been proposed to deal with the issue [10, 13, 37]. Specifically, for RELU-activation networks, Kaiming initialization of parameters [13] stabilizes the variances, keeping them constant through the layers. In the context of ResNets, the variance of the input and output of residual blocks will remain identical with this initialization, *i.e.* $\mathrm{Var}(\mathcal{F}_l(\boldsymbol{x}_{l-1})) = \mathrm{Var}(\boldsymbol{x}_{l-1})$ (where $\mathrm{Var}(x)$ refers to the vector of variances of the individual components of $x$).

However, this leads to a new problem in a plain ResNet: as explained in [40], the correlations between $\boldsymbol{x}_l$ and $\mathcal{F}_l(\boldsymbol{x}_{l-1})$ are small (also see Appendix A.1), and at each residual connection we get

$$\mathrm{Var}(\boldsymbol{x}_l) = \mathrm{Var}(\boldsymbol{x}_{l-1} + \mathcal{F}_l(\boldsymbol{x}_{l-1})) \approx \mathrm{Var}(\boldsymbol{x}_{l-1}) + \mathrm{Var}(\mathcal{F}_l(\boldsymbol{x}_{l-1})) = 2\mathrm{Var}(\boldsymbol{x}_{l-1}).$$

Thus the variance doubles at each residual block, and increases exponentially with the number of residual blocks, leading to an identical problem in the backward pass during training. Although [40] do provide a solution, this sometimes comes at the cost of a minor loss of performance.

## 3.2 The "Dead ReLU" Problem in Plain Networks

In a ReLU-activation network, some fraction of neurons never get activated, and always produce zero output regardless of the input. This "Dead Relu" problem is generally believed to be related to improper initialization with large gradients. Instead, we have the following proposition: for a deep rectifier neural network, even with careful initialization using current techniques [10, 13], dead ReLUs occur in significant numbers right at the outset after initialization and never recover.

Consider one linear layer: $\boldsymbol{y}_k = \boldsymbol{W}_k \boldsymbol{x_k} + \boldsymbol{b}_k$, $\boldsymbol{x}_k = \mathrm{ReLU}(\boldsymbol{y}_{k-1})$. Suppose $\boldsymbol{W}_k$ are initialized with He Initialization [13]: $w_{ij}^k \sim \mathcal{N}(0, 2/d)$, $\boldsymbol{b}_k = 0$. Assume the elements of $\boldsymbol{x}_k$ have expectation $\mathrm{c}_k$.

$$\mathbb{E}([\boldsymbol{y}_k]_i) = \mathbb{E}(\sum_{j=1}^{d} w_{ij}^k [\boldsymbol{x}_k]_j) = \mathrm{c}_k W_i^k, \ \ \text{where} \ \ W_i^k = \sum_{j=1}^{d} w_{ij}, \tag{1}$$

where the notation $[\boldsymbol{v}]_i$ for any vector $\boldsymbol{v}$ represents the $i^{\text{th}}$ component of the vector. Since $\boldsymbol{x}_k$ comes from a ReLU layer, $c_k$ is positive (or non-negative). Once initialized, the weights are no longer random. $W_i^k$ is a hence a single sample drawn from a Gaussian: $\mathcal{N}(0, 2)$, with a significant probability of being negative or even highly negative. Consequently, the PDF of many components of $\boldsymbol{y}_k$ will be centered around a large negative value, as a result of which that component will be wiped out by the subsequent ReLU activation (and never recover since the derivatives too will now be 0).

A consequence of the dead ReLU problem is that a very large fraction of neurons in a ReLU network can become ineffective at the very outset of training. For instance in our simulations in Appendix A.2, nearly 40% of the neurons in the $20^{\text{th}}$ layer of a deep ReLU-activation network are potentially dead or otherwise unable to model non-linearity at initialization.

# 4 RescaleNet

We now introduce *RescaleNet*, our variation on the residual architecture that does not require normalization. First we introduce the basic formalism, and then discuss various extensions that all provide additional benefits.

## 4.1 RescaleNet for Residual Learning

The problem of vanishing/exploding values is, arguably, the biggest challenge in training non-normalized networks. Due to the additive combination of residual blocks, $\boldsymbol{x}_k = \boldsymbol{x}_{k-1} + \mathcal{F}_k(\boldsymbol{x}_{k-1})$,

the problem of exploding variance stated in Section 3 cannot be resolved through initialization alone, and the actual computations in the model must be modified. Instead, we propose the following re-scaling:

$$\boldsymbol{x}_k = \alpha_k \boldsymbol{x}_{k-1} + \beta_k \mathcal{F}_k(\boldsymbol{x}_{k-1}), \tag{2}$$

where $\alpha_k^2 + \beta_k^2 = 1$.

Suppose $\mathcal{F}_k$ includes no normalization and the weights are initialized by [13], the inputs and outputs of $\mathcal{F}_k$ have approximately equal variances: $\text{Var}[\mathcal{F}_k(\boldsymbol{x}_{k-1})] \approx \text{Var}[\boldsymbol{x}_{k-1}]$. Assume (following Section 3.1) that there is little linear correlation between $\mathcal{F}_k(\boldsymbol{x}_{k-1})$ and $\boldsymbol{x}_{k-1}$. We have:

$$\text{Var}[\boldsymbol{x}_k] = \alpha_k^2 \text{Var}[\boldsymbol{x}_{k-1}] + \beta_k^2 \text{Var}[\mathcal{F}_k(\boldsymbol{x}_{k-1})] = \text{Var}[\boldsymbol{x}_{k-1}], \tag{3}$$

ensuring stable variances, provided $\alpha_k$ and $\beta_k$ are properly set.

Expanding the recursion of Equation 2 recursively, assuming $L$ residual blocks in all, we can write:

$$\boldsymbol{x}_L = (\prod_{i=1}^{L} \alpha_i)\boldsymbol{x}_0 + \sum_{k=1}^{L} \beta_k \prod_{i=k+1}^{L} \alpha_i \mathcal{F}_k(\boldsymbol{x}_{k-1}). \tag{4}$$

The optimal coefficients should ensure that different residual blocks have the same weights:

$$\forall k \neq k', \beta_k \prod_{i=k+1}^{L} \alpha_i = \beta_{k'} \prod_{i=k'+1}^{L} \alpha_i. \tag{5}$$

Solving Equation 5 with $\alpha_k^2 + \beta_k^2 = 1$, we get (see Appendix A.3): $\alpha_k = \sqrt{(k-1+c)/(k+c)}$, $\beta_k = 1/\sqrt{k+c}$, where $c$ is a hyper-parameter. To see which hyper-parameter $c$ is better, we re-write $\mathcal{F}_k(\boldsymbol{x}_{k-1})$ as $\mathcal{F}_k(\boldsymbol{x}_{k-1}, \Theta_k)$ where $\Theta_k$ is the parameter in this residual block:

$$\boldsymbol{x}_k = \sqrt{\frac{k-1+c}{k+c}}\boldsymbol{x}_{k-1} + \sqrt{\frac{1}{k+c}}\mathcal{F}_k(\boldsymbol{x}_{k-1}, \Theta_k) \tag{6}$$

It is sufficient to overcome the problem of unstable variances and gradients in un-normalized nets with the basic formalism of Equation 6. However, it it does not necessarily mean that the network is trainable. If the gradient is too large, the output of the network may still overflow. Consider a loss function $\ell(\boldsymbol{w})$ which is being minimized w.r.t. a parameter $\boldsymbol{w}$ using gradient decent. Let $\eta$ be the learning rate, we have $\boldsymbol{w}_{t+1} = \boldsymbol{w}_t - \eta \nabla \ell(\boldsymbol{w}_t)$. After one update step, the change to the loss is:

$$\Delta \ell = \ell(\boldsymbol{w}_{t+1}) - \ell(\boldsymbol{w}_t) = \ell(\boldsymbol{w}_t - \eta \nabla \ell(\boldsymbol{w}_t)) - \ell(\boldsymbol{w}) = -\eta \|\nabla \ell(\boldsymbol{w}_t)\|_2^2 + O(\eta^2). \tag{7}$$

If then length of the gradient is too large, we need a correspondingly small learning rate $\eta$. Otherwise, a big change to the loss would make it overflow.

In our network (Equation 6), let $\dfrac{\partial \ell}{\partial \boldsymbol{x}_k}$ be the derivative of the loss w.r.t $\boldsymbol{x}_k$ and $\dfrac{\partial \mathcal{F}_k}{\partial \Theta_k}$ be the derivative of the residual branch $\mathcal{F}_k$ w.r.t to parameter $\Theta_k$. We can assume the derivatives maintain a similar magnitude in different layers by using He Initialization and our rescaling scheme. The square summation of all parameters' gradient is:

$$\|\nabla \ell\|_2^2 = \sum_{k=1}^{L} \left( \frac{\partial \ell}{\partial \boldsymbol{x}_k} \cdot \sqrt{\frac{1}{k+c}} \frac{\partial \mathcal{F}_k}{\partial \Theta_k} \right)^2 \approx \sum_{k=1}^{L} \frac{1}{k+c} \| \frac{\partial \ell}{\partial \boldsymbol{x}_k} \|_2^2 \| \frac{\partial \mathcal{F}_k}{\partial \Theta_k} \|_F^2 = O(\sum_{k=1}^{L} \frac{1}{k+c}) \tag{8}$$

The hyper-parameter should be neither too small nor too large. For example, a small hyper-parameter, say $c = 1$, make gradient unbounded: $\|\nabla \ell\|_2^2 = O(\ln L)$. We choose $c = L$, the number of residual blocks, which makes the square summation of all parameters' gradient $\|\nabla \ell\|_2^2$ less dependent on the network depth. The residual connection of Equation 6 is now changed to to:

$$\boldsymbol{x}_k = \sqrt{\frac{k-1+L}{k+L}}\boldsymbol{x}_{k-1} + \sqrt{\frac{1}{k+L}}\mathcal{F}_k(\boldsymbol{x}_{k-1}, \Theta_k) \tag{9}$$

We will refer to this network formalism as *RescaleNet*. The basic RescaleNet formalism of Equation 9 is sufficient to overcome the problem of unstable variances and gradients in un-normalized nets. However its performance can be further improved through the extensions given below.

### 4.2 Scalar multipliers

The choice of $c$ influences how the weight given to different layers. A smaller $c$, say $c = 1$, gives higher weight to the deeper residual branches. A larger $c$, gives higher weight to the shallow layers (usually there are still a few layers between the input and the residual blocks). However, the importance of shallow and deep layers change during training. In the early stage of training, the shallow layers learns most and the deep layers do not learn much [39]. Without good features from shallow layers, deep layers cannot learn well. As the training going, deeper layers turn more important since they have many more dimensions/parameters and are more close to the loss function.

To match the training dynamic, we select a relatively big $c = L$ and include a learnable multiplier initialized to 1 at each residual block. Let $m_k$ be the learnable multiplier. The network can en-weight deeper layers by learning a larger multiplier. The final residual learning equation is:

$$\boldsymbol{x}_k = \sqrt{\frac{k-1+L}{k+L}}\boldsymbol{x}_{k-1} + \frac{m_k}{\sqrt{L}}\mathcal{F}_k(\boldsymbol{x}_{k-1}) \tag{10}$$

By default, $m_k$ should be a vector (similar to the weights of BN). However, we find scalar multipliers are better [2]. Scalar multipliers can have more updates than vector multipliers (the gradient of scalar multipliers are accumulated from all dimensions). The final learned scalar multipliers can be up to $O(L)$, however the final learned vector multipliers ranges from 1 to 4, which may not be big enough to up-weight deeper layers.

A larger multipliers may break our previous theories about stable variance. However, we find the convolution weights also shrink during training due to weight decay. The decrease in weights and increase in learnable multipliers compensate for each other, making the variance of residual block outputs in an acceptable range. A detailed study about such dynamic could be a good future work.

### 4.3 Bias Initialization

In models that include normalization layers, the preceding linear layer usually has no bias since it may be canceled by the mean reduction operation of normalization. When normalization is removed from the network, we must consider a proper setting for the bias term. The common setting is to put bias after the matrix multiplication, i.e. $\boldsymbol{y} = \boldsymbol{Wx} + \boldsymbol{b}$, and initialize the bias as zero. However, as discussed in Section 3.2 this can result in a large fraction of dead ReLUs, effectively losing much of the modelling capacity of the network.

Instead, we apply the bias before the matrix multiplication, i.e. $\boldsymbol{y} = \boldsymbol{W}(\boldsymbol{x} + \boldsymbol{b})$, and initialize the bias in a data-dependent manner (in ReLU networks), as the negative of the mean of the first mini-batch of data used during training. We find bias before the matrix multiplication is much easier to optimize. Data-dependent initialization can greatly mitigate the dead ReLU problem: all neurons have both positive and negative outputs since the mean of the first batch of data is subtracted.

## 5 Experiments

### 5.1 ImageNet Classification

We experiment in the ImageNet classification dataset [8]. The dataset contains 128k training images and 50k validation images that are labeled with 1000 categories.

**Implementation details.** We follow the official PyTorch implementations [25]. During training, we adopt random resized crop with a $224 \times 224$ crop size, and random horizontal flip for data augmentation. We use SGD to train the models for 100 epochs. We use a weight decay of 0.0001 for all weight layers, and no weight decay for the bias and multipliers. We report the top-1 classification error on the $224 \times 224$ center-crop in the validation set. All results are averaged over 5 runs.

The default setting is to train the model with a batch size of 256 and an initial learning rate of 0.1. The learning rate is decreased at 30, 60, 90 epochs. To accelerate training, we also try to train the model with a batch size of 1024 and increase the learning rate by 4 times. To match the performance

of batch size 256, we use gradual warmup [12] for 5 extra epochs. The performance with batch size 1024 is about 0.1% lower than with batch size 256. Unless otherwise stated, we use batch size 1024.

**Ablation Study.** Our baseline RescaleNet model for ablation study includes all the components of Section 4.3: Equation 10 for residual connections (with hyper-parameter $c$ set to $L$), the data-dependent bias, and a learnable *scalar* multiplier. Table 1 shows the following ablation studies:

1. The choice of hyper-parameter $c$: as discussed in Section 4.3, $c$ controls the relative contributions of deep and shallow layers. We compare $c = L$ (baseline) with $c = 1$ and $c = L^2$. As show in Table 1 (a), the results are not sensitive to $c$, and the baseline setting is slightly better (but has a theoretical guarantee!).

2. The effectiveness of the multiplier: we compare three variants of the multiplier: no multiplier, a scalar multiplier (baseline) and a vector multiplier.

3. Bias: we compare three cases – bias applied prior to multiplication by weights (Pre bias), and initialized in a data-dependent manner (Data Init) (Section 4.3, baseline), Pre bias initialized to 0 (Zero Init), and a data-initialized bias applied after weight multiplication (Post bias).

| (a) Hyper-parameter $c$ | | (b) Multiplier Setting | | (c) Bias Setting | |
|---|---|---|---|---|---|
| Method | Accuracy | Method | Accuracy | Method | Accuracy |
| $c = L$ | **76.6%** | Scale multiplier | **76.6%** | Pre bias + Data Init | **76.6%** |
| $c = 1$ | 76.4% | No multiplier | 74.2% | Pre bias + Zero Init | 76.4% |
| $c = L^2$ | 76.3% | Vector multiplier | 74.8% | Post bias + Data Init | 76.1% |

Table 1: Ablation Study on ImageNet with ResNet backbone (first line is the baseline)

**Comparison with non-normalized models.** We compare our method with two related studies on non-normalized models: Fixup-Init [40] and SkipInit [7]. Table 2 shows ImageNet Results with the ResNet50 backbone. Results without citations are based on our implementation. Since normalization has a strong regularisation effect, non-normalized models suffer from overfitting. We consider two kinds of extra regularisation: 1) Mixup data augmentation with a coefficient $\alpha = 0.7$. The coefficient is selected by cross-validation on Fixup-Init models [40]. 2) Adding spatial dropout with a drop rate[3] $p = 0.03$ to convolution layers in conv4_x and conv5_x of the ResNet50 backbone and dropout with a drop rate $p = 0.3$ before the last linear layer. RescaleNet surpasses other non-normalized models by a significant margin. One thing worth mentioning is that **RescaleNet is able to train well with the default ResNet50 training setting**. However, the other two methods require special training settings: Fixup-Init reduces the learning rate of some parameters by 10 times; SkipInit reduces the learning by a factor of 2 every 5 epochs in the middle of training.

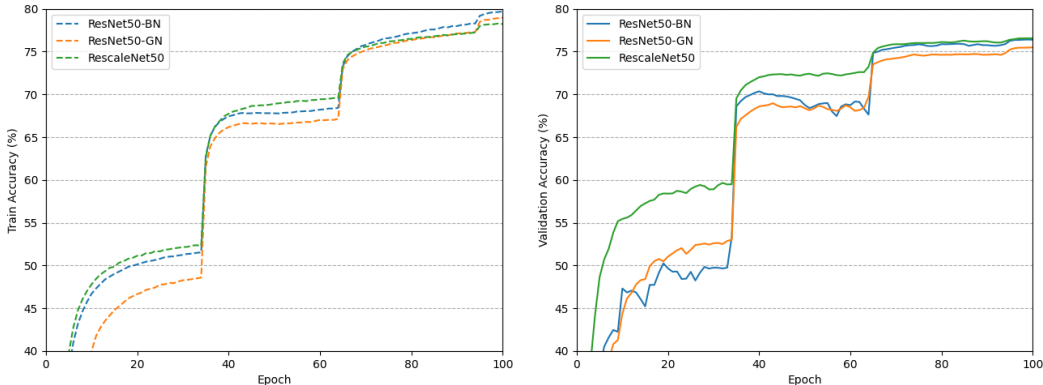

Figure 1: Training (left) and validation (right) accuracy(%) on ImageNet using ResNet50

**Comparison with normalized models.** We choose RescaleNet + Dropout as our baseline model and compare with some normalization methods. Table 3 shows the results on ImageNet validation dataset.

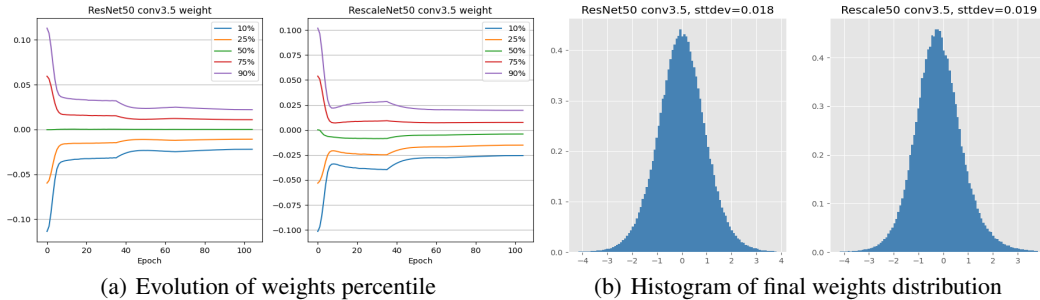

(a) Evolution of weights percentile      (b) Histogram of final weights distribution

Figure 2: Visualization of weights from ResNet and RescaleNet during and after training

| Method | Regularization | Accuracy |
|---|---|---|
| Fixup-Init | None [40] | 72.4% |
| | Mixup [40] | 76.0% |
| | Dropout | 75.5% |
| SkipInit | None [7] | 74.9% |
| | Dropout [7] | 75.6% |
| RescaleNet | None | 74.3% |
| | Mixup | 76.4% |
| | Dropout | **76.6%** |

Table 2: ResNet50 validation accuracies on ImageNet for non-normalization methods.

| Method | Accuracy(%) |
|---|---|
| Batch Normalization [17] | 76.4 |
| Layer Normalization [2] | 74.7 |
| Instance Normalization [33] | 71.6 |
| Group Normalization [36] | 75.9 |
| Switchable Normalization [21] | 76.9 |
| RescaleNet | 76.58±0.08 |
| Filter Response Norm [31] | 77.2 |
| RescaleNet + Cosine LR | **77.2±0.06** |

Table 3: ResNet50 Validation accuracies on ImageNet compared with normalization methods.

Since Filter Response Normalization (FRN) uses cosine learning rate decay [12] for optimization, we also include the corresponding result for RescaleNet. Our method's performance is better than the traditional normalization method (BN, GN, etc), and very close to the performance of the state of the art methods (SN, FRN).

Figure 1 shows the accuracy curves. RescaleNet has a rapid convergence in the first 60 epochs, but BN/GN models catch up a little in the last 40 epochs. Figure 2 (a) shows how the weights percentile in Rescale50 (RescaleNet version of ResNet50) and ResNet50-BN models vary during training (conv3.5 is the last convolution in the $3^{rd}$ stage). Figure 2 (b) shows the histograms of the weights. The weights are standardized and we print the initial standard deviation of the weights on top of the figures. As shown in the figure, both weights in Rescale50 and ResNet50-BN models have a healthy distribution.

When the batch size becomes smaller, the performance of RescaleNet does not drop since it does not require batch statistics. If the batch size is 1, we can update the parameters every 256 backward passes, which is equivalent to training the model with a batch size of 256. However, the validation accuracy of Switchable Norm would drop by 1.3% when the batch size is reduced to 1.

**Performance on more architectures.** To validate the generalization ability of the proposed method, we compare RescaleNet with the corresponding normalized model on a series of architectures. The results are shown in Table 4. Though the re-scaling itself does not apply to conventional nets without residual connections, the pre-weight bias strategy does apply and also proves to be very effective. Table 5 shows the result on VGG-19. By only applying the pre-weight bias strategy, the non-normalized model can almost match the performance of the BN model.

| Architecture | BN model(%) | Rescale model(%) |
|---|---|---|
| ResNet101 | 77.4 | 77.53±0.08 |
| ResNeXt101 | 79.3 | 79.45±0.05 |
| SE-ResNet50 | 76.7 | 77.22±0.11 |

Table 4: Validation accuracy of BN and RescaleNet models on ImageNet.

| Architecture | Accuracy(%) |
|---|---|
| vanilla VGG 19 | 74.4 |
| VGG 19 + BN | 75.1 |
| VGG 19 + Prebias | 75.0±0.12 |

Table 5: VGG19 validation accuracies on ImageNet.

To seek the best possible performance of our method, we also train our method with several training procedure refinements. Following [16], we use the ResNet-D architecture, apply label smoothing and mixup in addition to the dropout regularization as discussed, and train for 200 epochs with warmup. We achieve a top-1 accuracy of **78.93%** on the ImageNet validation dataset. The corresponding BN model achieves a top-1 accuracy of 79.15%, i.e., only 0.22% improvement with BN enabled.

## 5.2 Experiments on More Tasks

**Object Detection and Segmentation.** These vision tasks take higher-resolution images as inputs, thus the batch size is very small (2 or 4 images per GPU). We adopt Mask R-CNN with a Feature Pyramid Network (FPN) backbone as the detection model, and compare the RescaleNet backbone with the corresponding BN/GN backbone. Our codes are based on the official PyTorch Implementation [27]. The models are trained in the COCO train2017 set and evaluated on the COCO val2017 set. We report the standard COCO metrics of Average Precision (AP), AP50 , and AP75 , for bounding box detection ($AP^{bbox}$) and instance segmentation ($AP^{mask}$). All models are trained for 180k iterations with a batch size of 16 (8 GPUs, 2 images per GPU).

Following [14], the BN layer is turned into a linear layer: $y = \gamma(x - \mu)/\sigma + \beta$ since the small batch size will decrease the performance if batch normalization is performed. The parameters of the backbones are initialized from the corresponding ImageNet classification pre-trained models. For fair comparisons, the pre-trained models for different methods (BN/GN and Rescale) have close performance on the ImageNet validation dataset. We apply pre-bias to all convolution layers and linear layers in the heads. Following [36], we use the 4conv1fc head indead of 2fc head. As shown in Table 6, RescaleNet model's performance is always better than the BN/GN model.

| Model | $AP^{bbox}$ | $AP^{bbox}_{50}$ | $AP^{bbox}_{75}$ | $AP^{mask}$ | $AP^{mask}_{50}$ | $AP^{mask}_{75}$ |
|---|---|---|---|---|---|---|
| ResNet50 BN 2X | 38.6 | 59.8 | 42.1 | 34.5 | 56.4 | 36.3 |
| ResNet50 GN 2X | 40.3 | 61.0 | 44.0 | 35.7 | 57.9 | 37.7 |
| RescaleNet50 2X | 40.4 | 60.8 | 44.1 | 35.9 | 57.7 | 37.9 |
| ResNet50GN 3X | 40.8 | 61.6 | 44.4 | 36.1 | 58.5 | 38.2 |
| RescaleNet50 3X | 41.0 | 61.6 | 45.1 | 36.9 | 58.8 | 39.6 |
| ResNet101 BN 2X | 40.9 | 61.9 | 44.8 | 36.4 | 58.5 | 38.7 |
| ResNet101 GN 2X | 41.8 | 62.5 | 45.1 | 36.8 | 59.2 | 39.0 |
| RescaleNet 101 2X | 41.8 | 62.7 | 45.2 | 37.0 | 59.2 | 39.3 |

Table 6: Detection and segmentation results in COCO using Mask R-CNN and FPN.

**Video classification.** We further evaluate anther vision task: video classification on the Kinetics dataset. We adopt the Inflated 3D ResNet50 [5]. In training, we sample a 32-frame clip with a stride of 4 frames from each video. In evaluation, we sample 10 clips uniformly, and the final prediction is the averaged softmax scores of all clips. All models are trained in the same setting as in [35] for 100 epochs with a batch size of 64 (8 GPUs, 8 videos per GPU). Table 7 shows the results.

| method | accuracy |
|---|---|
| BN [36] | 73.3% |
| GN [36] | 73.0% |
| Rescale | **73.7%** |

Table 7: Top-1 accuracy on Kinetics Validation.

| method | BLEU |
|---|---|
| Layer Norm [40] | 34.2% |
| Fixup Init [40] | 34.5% |
| Rescale | **35.0%** |

Table 8: Machine translation on IWSLT DE-EN.

| method | BLEU |
|---|---|
| Layer Norm [24] | 29.3% |
| Fixup Init [40] | 29.3% |
| Rescale | **29.6%** |

Table 9: Machine translation on IWSLT DE-EN.

**Machine Translation.** Finally, we evaluate our method in transformer [34], a state of the art architecture in language models. We remove all Layer Norms in transformer, and re-scale all residual connections using methods discussed in Section 4. Following [40], we choose the machine translation task and test on two datasets: IWSLT 2014 German-English (de-en) [6] and WMT 2016 English-German (en-de) [24]. We use the fairseq library [23] as our code base, and follow the same settings as [40]. Table 8 and 9 show the results on two datasets. A higher BLEU is better.

# 6 Conclusion

In this work, we investigated how to train deep neural networks without normalization layers and without performance degradation. We discussed the exploding variance and "Dead ReLU" problems in neural networks. We then proposed RescaleNet, a variation on the residual architecture that does not require normalization. We further proposed several simple extensions to improve its performance. We demonstrated the effectiveness of our method on a wide range of tasks: image classification, object detection and segmentation, video classification, and machine translation. Our method is competitive to normalization methods, thus investigating RescaleNet facilitates the theories of normalization. In future work, we will have more detailed comparisons between RescaleNet and normalization models (BN) in many aspects, for example loss landscape and training dynamics.

## Acknowledgement

The authors thank Lingxiao Zhao, Haoqi Hu from Carnegie Mellon University and Chengming Xu from Fudan University for discussions on the paper. This work is generously supported in part by NSFC Project U1611461, Shanghai Municipal Science and Technology Major Project (2018SHZDZX01), and Shanghai Research and Innovation Functional Program (17DZ2260900).

## Broader Impact

Our research studies a lower level problem of deep learning, i.e., the architecture of neural networks. Researchers who are interested in the functionalities of normalization or more generally, the training of deep neural networks, may get some insights from our paper. Further, our method can be used to train neural networks with small batch size, thus researchers with limited computation resources may benefit from this.

Since our research does not involve any specific high-level AI applications, we do not think there would be any people being put at disadvantage from this research to the best of our knowledge. The meaning of our research more lies in terms of theory, thus we cannot see any bad consequences of a failure of the system.

We follow the most common settings in the deep learning community to process data and the task/method does not leverage biases in the data.

## Footnotes

[2]Zhang *et al.* [40] also used scalar multipliers, but they did not emphasize scalar multipliers are different from, and much better than vector multipliers.

[3]We choose the dropout rate heuristically. The regular batch size for BN is 32 images per GPU, thus the variance of noise introduced from batch statistics is about $1/32 \approx 0.03$ of the batch variance.

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
