[Supplementary Material]

# 7 Appendix

**A.1 The input and output of the residual block are weakly correlated** With the formulation of residual connections, we have: $\text{Var}([\boldsymbol{x}_{k+1}]_i) = \text{Var}([\boldsymbol{x}_k]_i) + \text{Var}([\mathcal{F}_k(\boldsymbol{x}_k)]_i) + \text{Cov}([\boldsymbol{x}_k]_i, [\mathcal{F}_k(\boldsymbol{x}_k)]_i)$. With He initialization [13] , the output variance of $\mathcal{F}_k$ can be approximately equal to the input variance: $\text{Var}([\mathcal{F}_k(\boldsymbol{x}_k)]_i) \approx \text{Var}([\boldsymbol{x}_k]_i)$.

To evaluate the covariance term, $\text{Cov}([\boldsymbol{x}_k]_i, [\mathcal{F}_k(\boldsymbol{x}_k)]_i)$, we assume that any two coordinates in $\boldsymbol{x}_k$ are uncorrelated. With this assumption, the covariance term is about $O(1/\sqrt{d})$ small. We only need to prove the one layer case: $\mathcal{F}(\boldsymbol{x}) = \boldsymbol{W}\boldsymbol{x}$. The elements of $\boldsymbol{W}$ can be sampled from Gaussian: $w_{ij} \sim \mathcal{N}(0, 1/d)$ to match the input/output variance.

$$\text{Cov}([\boldsymbol{x}_k]_i, [\mathcal{F}_k(\boldsymbol{x}_k)]_i) = \text{Cov}([\boldsymbol{x}_k]_i, \sum_{j=1}^{d} w_{ij}[\boldsymbol{x}_k]_j) = \sum_{j=1}^{d} w_{ij}\text{Cov}([\boldsymbol{x}_k]_i, [\boldsymbol{x}_k]_j) = w_{ii}\text{Var}([\boldsymbol{x}_k]_i). \quad (11)$$

The last equation holds since we assume any two coordinates in $\boldsymbol{x}_k$ are uncorrelated. By Eq 11, $\text{Var}([(\boldsymbol{x}_{k+1})]_i) \approx (2 + w_{ii})\text{Var}([\boldsymbol{x}_k]_i)$. Recall that $w_{ii} \sim \mathcal{N}(0, 1/d)$, with a probability at least $1 - \exp(-d/4)$, $2 + w_{ii} > 2 - 1/\sqrt{2}$. In common ResNet, $d \geq 32$, which makes the residual block output variance increases exponentially with a very high probability.

If there are multiple layers in the residual block $\mathcal{F}_k$, ReLU activations would further decrease (at least not increase) the correlation between $[\boldsymbol{x}]_i$ and $[\mathcal{F}_k(\boldsymbol{x}_k)]_i$. We have have the following claim:

**Proposition 1.** Correlations between $\boldsymbol{x}_k$ and $\mathcal{F}_k(\boldsymbol{x_k})$ are about $O(\sqrt{d})$ small.

(a) Percentage of four categories in a 50 layer network. (b) Histogram of two neurons' response in the $5^{\text{th}}$ layer.

Figure 3: Visualization of the Dead ReLU problem immediate after initialization.

**A.2 The dead ReLU problem** To verify the extent of the dead ReLU problem, we ran an experiment on a deep fully-connected ReLU-activation network. The input dimension, as well was the output dimension of all layers was 1024. We used He initialization for all fully-connected layers. The inputs were drawn from a standard Gaussian: $\boldsymbol{x} \in \mathbb{R}^{1024} \sim \mathcal{N}(0, 1)$. The outputs of the $k^{\text{th}}$ layer before activation are:

$$\boldsymbol{x}_k = \boldsymbol{W}_k\Big(\text{ReLU}\big(\boldsymbol{W}_{k-1}\cdots\text{ReLU}(\boldsymbol{W}_1\boldsymbol{x})\big)\Big) \in \mathbb{R}^{1024}. \quad (12)$$

We generate 4096 pieces of inputs from standard Gaussian, and examine each layer's outputs of the network directly after initialization, i.e., no training process. Let $[\boldsymbol{x}_k]_j$ be the $j^{\text{th}}$ neurons in the $k^{\text{th}}$ layer, and it has 4096 outputs from the 4096 inputs. We divide all neurons into three disjoint categories:

`all positive` : If all 4096 entries of $[\boldsymbol{x}_k]_j$ are positive,

`all negative` : If all 4096 entries of $[\boldsymbol{x}_k]_j$ are negative,

`nonlinear regime` : If there are both positive and negative entries in $[\boldsymbol{x}_k]_j$.

Any neuron in either `all positive` category or `all negative` category is labelled as being in the `linear regime` since the ReLU activation does not work for this neuron.

The percentage of neurons that fall into each the four categories against the layer index is shown in Figure 3 (a). The dead neurons in the $10^{\text{th}}$ layer have reached a proportion that cannot be ignored. About 40% of the neurons in the $20^{\text{th}}$ layer (e.g. VGG models) do not have the ability to capture non-linear relations, immediately after initialization. Figure 3 (b) shows the histogram of two neurons' response in the $5^{\text{th}}$ layer. The two neurons lose nonlinear representation ability immediate after initialization.

**A.3 Deriving the optimal weights**    Let $k'$ in Equation 5 be $k + 1$:

$$\beta_k \prod_{i=k+1}^{L} \alpha_i = \beta_{k+1} \prod_{i=k+1+1}^{L} \alpha_i \Longrightarrow \beta_k \alpha_{k+1} = \beta_{k+1}. \tag{13}$$

Recall $\alpha_{k+1}^2 = 1 - \beta_{k+1}^2$, we have $\beta_k^2 (1 - \beta_{k+1}^2) = \beta_{k+1}^2$, which turns to be:

$$\frac{1}{\beta_{k+1}^2} - \frac{1}{\beta_k^2} = 1 \Longrightarrow \frac{1}{\beta_k^2} = k + c. \tag{14}$$

Here $c$ is a constant.

**A.4 Fixed residual scaling**    This subsection was initially in the main body. However we feel that this part is a little complicated and the method in this part makes reviewers confusing. We feel that this subsection is not the nature of our proposed method and the improvement is a bit marginal. Therefore we decide to regard this part as an optional trick. and move it to the appendix to reflect the reviewer version of our paper.

Equation 16, combined with the initialization from [13], ensure the variance of the input and output of any residual block remain identical, and consequently, the norms of the gradient of the loss w.r.t. the output and input of each residuals section are equal in length as well (since gradient back propagation through the block mirrors the forward propagation). However, since the $1/\sqrt{k + L}$ term in Equation 16 is different for each residual block, the norm of the gradient through each residual block $\mathcal{F}_k$ will decrease with the depth of the block (while the gradient of the main signal will increase).

Consider a loss function $\ell(\boldsymbol{w})$ which is being minimized w.r.t. a parameter $\boldsymbol{w}$ using gradient decent. Let $\eta$ be the learning rate. After one update step, the change to the loss is:

$$\Delta \ell = \ell(\boldsymbol{w} - \eta \frac{\partial \ell}{\partial \boldsymbol{w}}) - \ell(\boldsymbol{w}) = -\eta \| \frac{\partial \ell}{\partial \boldsymbol{w}} \|_2^2 + O(\eta^2). \tag{15}$$

Assuming $\eta$ is small and the $O(\eta^2)$ terms can be ignored, we see that the decrease in loss is proportional to the squared length of the gradient. For the RescaleNet of Equation 6, therefore, the contribution from the last residual block to the loss reduction (scaled by $1/\sqrt{2L}$) is only half of the contribution from the first residual block (scaled by $1/\sqrt{L + 1}$).

This is a consequence of having variable weights for the residual blocks. However, we note that maintaining the variance of $\boldsymbol{x}_k$ can also be achieved if we *fix the weight of the residual block,* $\beta_k$, and let $\text{Var}(\mathcal{F}_k(\boldsymbol{x}_{k-1}))$ shrink instead. This is achieved through a modification of the He initialization, whereby we initialize all residual weights for $\mathcal{F}_k()$ by drawing from a Gaussian distribution $\mathcal{N}(0, \frac{2}{d}(\frac{L}{k+L})^{2/N})$, where $d$ is the number of weights for each neuron/filter, and $N$ is the number of layers in $\mathcal{F}_k()$. This ensures that $\text{Var}(\mathcal{F}_k(\boldsymbol{x}_{k-1})) = \frac{L}{k+L} \text{Var}(\boldsymbol{x}_{k-1})$. The residual connection of Equation 6 is now changed to to:

$$\boldsymbol{x}_k = \sqrt{\frac{k - 1 + L}{k + L}} \boldsymbol{x}_{k-1} + \frac{1}{\sqrt{L}} \mathcal{F}_k(\boldsymbol{x}_{k-1}) \tag{16}$$