[Reviews · NeurIPS 2020]

Review 1

Summary and Contributions: This paper investigates how to train deep neural networks without normalization layers and without performance degradation. Authors propose RescaleNet, a ResNet variant architecture that do not require normalizations. Authors show that by introducing scalar multiplier to the identity/residual connections, you can preserve the variance of the signal through the forward propagation in the networks. Authors perform an extensive empirical evaluation on various tasks (ImageNet, MS-COCO, Kinetics and WMT) and show that RescaleNet matches or outperforms its normalized counterpart.

Strengths: The main strength of this paper lies in its empirical evaluation. The paper proposes an impressive empirical. It shows on a wide variety of tasks that RescaleNet can matches normalized Deep Networks. They also show that their approach outperforms other normalization-less methods such as FixUp and SkipInit. The paper also performs an ablation study to show the impact of the different design choice and hyperparameter. Assumption made in the analysis part should be more explicit in the main paper and it would be nice to discuss how realistic they are. In particular, the analysis section assumes that two coordinates in x_k are are uncorrelated.

Weaknesses: Assumption made in the analysis part should be more explicit in the main paper and it would be nice to discuss how realistic they are. In particular, the analysis section assumes that two coordinates in x_k are are uncorrelated.The claims and the method appear correct to me.

Correctness: The claims and the method appear correct to me.

Clarity: The paper is clear an easy to follow.

Relation to Prior Work: Relation to prior work is clear written.

Reproducibility: Yes

Additional Feedback: Update after rebuttal: Thanks for writing the rebuttal. I appreciate the authors trying to address some of the concerns. I believe the paper is interesting and and I will on keeping my original positive rating in favor of acceptance.


Review 2

Summary and Contributions: The authors propose a family of architectures, called RescaleNets, that are variants of residual networks. The modified architecture carefully weights the residual connection and backbone of each block to ensure that the variance of the signal stays constant and each layer contributes equally to the output. Notably RescaleNets do not include any normalization layers. The authors augment this architecture with: 2) A reparameterization of the residual blocks to additionally equalize the contribution of each layer to the change in loss after a single step of gradient descent. 3) A learned independent scalar or vector multiplier on the residual blocks to fine-tune their contribution to the loss. 4) A reparameterization of the affine transformations to ensure that neurons don’t die due to the ReLU activations. The authors present a wide array of experiments on ImageNet as well as object detection, video classification, and machine translation. In many cases, RescaleNet fares better than normalization based architectures.

Strengths: Overall I liked this paper a lot! 1) I thought the proposed changes to ResNet were well motivated and the performance was impressive. 2) It is nice to see such strong results from what is essentially careful initialization. Removing normalization layers seems like it would be a very nice win for making models simpler and easier to reason about. 3) I appreciated the ablation study showing various choices of c, the multiplier, and the reparameterization of the bias. 4) Overall, I think that the authors make a very compelling case for the reparameterization of the bias which seems like it should be widely used. 5) I appreciate that the authors showed performance outside of ImageNet. 6) The authors included code to reproduce the imagenet results from their experiments.

Weaknesses: Things I would have liked to see / am curious about in order of decreasing importance: 1) It would be nice to compare against batchnorm + mixup training for 200 epochs. In the original Mixup paper, they observe only a 0.3% improvement at 90 epochs but this grows to 1.5% after 200 epochs of training. Does your method continue to improve as well? 2) Numerical evidence for some of the claims made in the theory section. It would have been nice to measure the variance of each layer and show that it is approximately constant and measure the contribution of each layer to the output to show that that is also constant. 3) I’m curious what the per-layer weights were after training. You say, The final learned scalar multipliers can be up to O(L), however the final learned vector multipliers ranges from 1 to 4, which may not be big enough 177 to en-weight deeper layers. But it would be nice to see this schematically. 4) How does subtracting the first batch means compare with subtracting the mean from a larger set of data? Can you plot the fraction of dead neurons against the size of the batch that is used to estimate the means? 5) Is there any benefit to including a trainable scalar multiplier on the backbone?

Correctness: As far as I can tell the results seem reasonable and correct. The proposed procedure is straightforward and uses very standard training procedures.

Clarity: 1) Overall I thought the exposition was clear, however there are some places in the text where I think clarity could be improved. I think the second section of 4.2 could use a bit more exposition. It took me a little while to figure out what was going on. It might be nice to explicitly write out how this change of parameterization affects the backward pass of the gradient computation. 2) I found the development of the architecture a little bit confusing. I would have described the final architecture as RescaleNet rather than describe RescaleNet + extensions. As a reader, why would I ever not use the extensions, especially since they seem to monotonically improve performance. 3) I think the term “en-weight” is not canonical. I would use something like “upweight”. 4) I found the sentence “We experiment in the ImageNet classification dataset” a bit awkward.

Relation to Prior Work: Overall I thought the authors did a reasonable job of citing prior work and choosing appropriate baselines.

Reproducibility: Yes

Additional Feedback: After rebuttal: Thanks to the authors for answering my questions, appreciate the responses. I found the plot of the learned scalars particularly interesting!


Review 3

Summary and Contributions: The authors develop a possible replacement for layer normalization methods on ResNets and achieve comparable accuracy with SOTA methods. In particular, they introduce RescaleNet, a scaled version of ResNet which applies a variety of rescalings to the standard ResNet construction to allow for better training by addressing the exploding variance and dead ReLU problems. The method is tested and applied for a variety of architectures.

Strengths: The core strength of this work are the empirical results, which showcase very competitive performance and seem to justify the central thesis of the paper. A key part of this is that the ideas can be applied to other models besides just ResNet across a variety of tasks, giving the indication that this method is fruitful. Furthermore, the paper also presents an interesting avenue of work by modifying both architecture and weight initialization. There seems to be a lot of promise in addressing these issues that the paper explores quite well.

Weaknesses: However, the paper still has a few flaws that should be addressed. First, the inclusion of scalar multipliers in section 4.3 seemingly contradicts the variance normalization induced by the alpha and beta in 4.1 and 4.2. In particular, having a >1 scalar term should still result in the “explosion” of the variance (from a theoretical lens) and seems to be rather important in achieving the desired results (Table 1b). Overall, this doesn’t sit well with the earlier sections. Second, what do the variances look like for some of the results. It appears that only 3 trials have been run (so maybe variance may be an inconclusive/incomplete measurement) but any indication of stability of the proposed method would be helpful.

Correctness: I have checked all of the derivations, and they all seem correct.

Clarity: The paper is well written. However, in table 2 the method is “fixup” not “mixup” (it’s the same lead author for both papers).

Relation to Prior Work: The work positions itself well in comparison with previous work and clearly delineates itself from previous papers.

Reproducibility: Yes

Additional Feedback: My score can be changed by addressing the weaknesses section in the rebuttal. Post-rebuttal: I don’t think that the authors addressed my concern (about the scaling term) quite well enough from a theoretical lens. This is because the authors implicitly admit that the scaling term should break the theoretical properties, although they do justify it with the fact that the weights tend to shrink as the training progresses. I’m a bit unsatisfied with this explanation, but I see the rationale. They do, however, adequately address my concerns about the results and stability.


Review 4

Summary and Contributions: 1. Motivated by exploding variance and DeadReLU problem, this paper proposed a rescale technique for the replacement of normalization techniques. 2. This paper is an extensive of research on training deep neural network without normalization. RESCALE method can achieve better performance than previous methods(training deep net without normalization). 3. The author derive a scaling parameter which can eliminate exploding variance and verify the effectiveness through experiments. Extensive ablation study on IMAGENET, COCO and WMT machine translation verify the technique soundness of RESCALE

Strengths: 1. Good explanation for previous research works, the theoretical derivation and experiment of RESCALE is quite easy to follow. 2. RESCALE can be a simple module to apply in practice due to the simplicity of implementation. 3. RESCALE combine the merits of normalization techniques and other good initialization techniques.

Weaknesses: 1. I am quite satisfied with the theory derivation and experiment. However, I am unclear the following details appeared in your uploaded code. The implementation of RESCALE is to multiply the residual branch with a depth dependent value. However, in your code, the initialization of convolution is dependent on the depth which has not been explained in the paper. Depth Scaled initialization is a technique beyond this paper. multiplier = (block_idx + 1) ** -(1/6) * max_block **(1/6) nn.init.normal_(m.weight, std = stddev * multiplier) Please correct me if I am wrong. 2. Currently, I am prone to weak accept considering the theory derivation is quite similar with previous works like skipinit and Fixup. I will change my score by considering the suggestion of other reviewers.

Correctness: The theory and experiment is correct.

Clarity: Well written paper with clear motivation, related work, theory prove and strong empirical results.

Relation to Prior Work: This paper is closely related with normalization, fixup initialization, rezero and skipinit. The relationship between related work has been discussed both theoretically and empirically.

Reproducibility: Yes

Additional Feedback:

[Author Response · NeurIPS 2020]

We thank all reviewers for their detailed suggestions and questions. Here are the responses to reviewers' questions.

**[R1]** *Assumption made in the analysis part should be more explicit. Discuss how realistic they are. Particularly, the*
*analysis section assumes that two coordinates in $x_k$ are uncorrelated.* **[R2]** *Numerical evidence for some of the claims.*
Thanks! We will write assumptions more explicitly in the final version. *Two coordinates in $x_k$ being uncorrelated* is a
widely used assumption [11]. We guess the reviewer may want to verify the assumption that any coordinate in $x_k$ and any
coordinate in $\mathcal{F}(x_k)$ are uncorrelated. If the two are uncorrelated, we have $\text{Var}[x_k] \approx \text{Var}[\alpha_k x_{x-1}] + \text{Var}[\beta_k \mathcal{F}(x_{k-1})]$
from Equation 2 in the submission paper. We define the relative error as:

$$\eta_k = \left| \frac{\text{Var}[\alpha_k x_{x-1}] + \text{Var}[\beta_k \mathcal{F}(x_{k-1})] - \text{Var}[x_k]}{\text{Var}[x_k]} \right| \times 100\% \tag{1}$$

Fig (a) shows the output variance $\text{Var}[x_k]$ and (b) shows the error $\eta_k$ using ImageNet data and Rescale101 model. The x

(a) y coordinate: residual block output variance      (b) y coordinate: relative error (%)

8
coordinate is the block index. The sudden drops in (a) are caused by downsample layers. Our assumptions are realistic.

**[R2]** *Compare against batchnorm + mixup training for 200 epochs; does the method continue to improve (with more*
*epochs)?* By training with mixup for 200 epochs, our baseline model (Table 3, 76.6%) achieve a **77.5%** top-1 accuracy.
The improvement (0.9%) is less than mixup (1.5%) since our method is not a regularization technique. We also compare
with "Bags of Tricks"[14] (line 253). The result shows our method can continue to improve with most training tricks.
**[R2]** *Schematically show the per-layer learned scalar multipliers:* Result from a ReScale50 checkpoint (16 scalars
from 16 residual blocks, from shallow ones to deep ones): 11.073, 10.339, 10.310, 14.467, 10.082, 13.121, 14.276,
16.991, 12.647, 15.192, 15.548, 17.261, 18.033, 27.445, 23.516, 27.258. More visualization results will be added.
**[R2]** *Subtract the first batch means vs. subtract the mean from a larger set of data*: we tried to subtract the mean from
more data, but find a batch size of 128 is all ready good enough: almost no neurons dies in the beginning of training. If
the model is too big to feed a batch of 128 data into GPUs, we can do this on CPU (just do one forward computation).
**[R2]** *Benefit of including a trainable scalar multiplier?* The benefits are two fold: 1) The convolution weights shrink in
the training due to weight decay (paper Fig 2 (a)), thus the large scalars can compensate for the shrink to keep the output
variance. 2) The learned scalars up-weights deeper blocks which have more parameters and higher-level features.

**[R3]** *Inclusion of scalar multipliers contradicts the variance normalization induced by the alpha and beta in 4.1 and*
*4.2; having a >1 scalar term should still result in the "explosion" of the variance* Thanks! Scalar multipliers are
initialized as one, thus are not a problem in the beginning of training. During training, the multipliers are updated to
larger values, meanwhile the weights also shrink due to weight decay (paper Fig 2(a)). Without scalar multipliers, the
output variance of each residual blocks would shrink as the weights shrink. Large multipliers can compensate for the
shrink. Experiments show that the variance of the last block output is always in a proper range in the final model.
**[R3]** *Variances of reported results*: Thanks! The method stability is indeed vital. We are repeating the reported
results with more trials. So far we have two results that may be of most interest to reviewers (mean±std from 6 runs):
RescaleNet50: 76.57%±0.084%, RescaleNet101: 77.55%±0.110%. More results will be added to the final version.

**[R4]** *Initialization of convolution depending on the depth was not been explained in the paper.* Thanks! We mentioned
this in line155-161. In Sec 4.1, we derived a $(k + L)^{-1/2}$ scaling coefficient for the $k^{\text{th}}$ block ($L$ is the number of
blocks). In line143-154 we mentioned this may not be good since the backward gradient would be scaled differently
by the layer-wise scaling. A fixed value scaling coefficient, say $L^{-1/2}$, is better. However, the output variance would
change if we only replace the scaling coefficient. Since the new coefficient is $L^{-1/2}(k + L)^{1/2}$ times larger, the output
variance of the residual branch should be $L^{1/2}(k + L)^{-1/2}$ times smaller. Note that there are three convolutions in
Bottleneck block, each convolution should be initialized with a $L^{1/6}(k + L)^{-1/6}$ times smaller standard deviation (In
our codes `block_idx` begins at $L$, so `block_idx+1` is $L + k$). We will write it more explicitly in the final version.
**[R4]** *Theory derivation is quite similar with previous works:* Thanks! Two differences from previous works' theories:
1) Fixup and Skip-Init are very similar to "zero-gamma" trick: their work use zeros to initialize the last layer in residual
branches, mimicking network that has less number of layers and are easier to train. Our method does not benefit
from zero initialization and all layers have non-zero outputs, which is more meaningful. It shows that deep networks
can be trained well without "zero-gamma". 2) Previous works mainly focus on how to make deep non-normalized
networks trainable. We further move on to how to train non-normalized models well (get the same performance as the
corresponding normalized models). For example, our method can also apply to non-residual network VGG with a very
competitive performance. Compared to previous works, our experiments are also more extensive.

[Meta-Review · NeurIPS 2020]

All four knowledgeable reviewers support acceptance. In particular, the reviewers found that the analysis was correct and the empirical evidence impressive. Therefore, I recommend accept. I recommend that the authors cite these earlier papers which propose a related scaling: Balduzzi, D., Frean, M., Leary, L., Lewis, J.P., Ma, K.W.D. and McWilliams, B., 2017. The shattered gradients problem: If resnets are the answer, then what is the question?. arXiv preprint arXiv:1702.08591. Gehring, J., Auli, M., Grangier, D., Yarats, D. and Dauphin, Y.N., 2017. Convolutional sequence to sequence learning. arXiv preprint arXiv:1705.03122.